# Sparse Image Synthesis via Joint Latent and RoI Flow

Ziteng Gao    Jay Zhangjie Wu    Mike Zheng Shou*

Show Lab, National University of Singapore

## Abstract

Natural images often exhibit underlying sparse structures, with information density varying significantly across different spatial locations. However, most generative models rely on dense grid-based pixels or latents, neglecting this inherent sparsity. In this paper, we explore modeling visual generation paradigm via sparse non-grid latent representations. Specifically, we design a sparse autoencoder that represents an image as a small number of latents with their positional properties (i.e., regions of interest, RoIs) with high reconstruction quality. We then explore training flow-matching transformers jointly on non-grid latents and RoI values. To the best knowledge, we are the first to address spatial sparsity using RoIs in generative process. Experimental results show that our sparse flow-based transformers have competitive performance compared with dense grid-based counterparts with significantly reduced lower compute, and reaches a competitive 2.76 FID with just 64 latents on class-conditional ImageNet $256 \times 256$ generation.

## 1 Introduction

Deep visual generative models have advanced significantly in recent years, achieving impressive visual quality on image [1, 2], video [3], and 3D domains [4]. Current visual generation pipelines typically start by encoding raw data (e.g., images) into compact latent representations with autoencoders, and then use diffusion, masking modeling, or autoregressive methods to generate such latents, as this pipeline exemplified by latent diffusion [1]. Flagship text-to-image models, e.g., Stable Diffusion [5] and FLUX [6], follow this line and compress spatial dimensions at typically $8\times$ factor, significantly lowering computational costs and modeling complexity in generative training.

Though being a core component of visual generation, autoencoders conventionally assume a grid-based space of latent structures with the uniform information density. However, natural images often exhibit highly non-uniform information density and require adaptive computation across spatial locations [7, 8]. For example, in a landscape image, the sky background occupies numerous pixels while being worth fewer latent units to reconstruct and generate. In contrast, intricate foreground objects may require more latents to capture their details. Existing visual generation pipelines fail to address this point, as they rely on dense uniform grid-based latent structures and cannot adaptively allocate more computation to intricate foregrounds.

This paper aims to study this point. First, we propose a *sparse visual autoencoder* that learns to compress an image into a set of sparse non-grid latents along with their positional property, i.e., region of interests (RoIs), and then recover image pixels from them. The RoIs explicitly characterize the spatial locations of the latents in bounding box formulation, and can be learned jointly with latents in an end-to-end manner by the plain reconstruction loss. The resulting sparse visual autoencoder reaches high compression rates by prioritizing latents to detailed regions while maintaining high reconstruction fidelity. Then, we design *sparse flow-based transformers* to generate latents and RoIs by modeling the joint flow of them with the velocity prediction in the denoising process [9, 10]. At

---

*The corresponding author.

39th Conference on Neural Information Processing Systems (NeurIPS 2025).

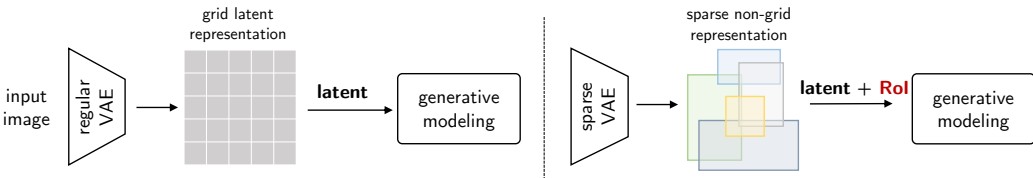

Figure 1: **Left:** conventional autoencoders encode pixels into latent grid representations. **Right:** our method encodes them into fewer non-grid latents with region of interests (RoIs).

every timestep, our model learns to estimate *both* latent and RoI velocity from initial noise to target samples. Divergent from prior grid-based latent approaches, our method dynamically adjusts latent spatial positions during sampling via ordinary differential equations (ODEs) at inference, allowing adaptive refinement of both content and spatial focus.

We show the feasibility of representing and generating images with sparse latents and RoIs on the challenge ImageNet benchmark. Our proposed sparse flow autoencoder, SF-VAE, can represent $256 \times 256$ images with just $64$ latents with $0.70$ reconstruction FID, or even down to $32$ latents with $1.70$ rFID. Then, the presented sparse flow-based transformers, SF-SiTs, have competitive performance on par with diffusion/flow-based grid-based transformers [11, 10]. The largest SF-SiT, XL variant, can reach $2.76$ FID with classifier-free guidance [12] on the class-conditional ImageNet generation benchmark with just $64$ latents.

## 2 Related Work

**Diffusion models.**    In recent years, generative models has been marked as a breakthrough in the field of visual synthesis [1, 13, 14]. Commercial systems like DALL-E [2] or FLUX [6] are typically rooted in denoising diffusion architectures. The seminal work, denoising diffusion probabilistic models [15], take the image generative process as a gradual denoising trajectory, iteratively refining pure noise into target images. Building on this foundation, subsequent advancements further accelerated and refined diffusion-based generation. Improved variants including [16, 17, 9] investigate the training and sampling trajectories, enabling high-quality results with fewer sampling steps. Latent diffusion models [1] democratize the high resolution image synthesis by operating in a compressed latent space with reduced computational costs. The following up work, including diffusion/flow-matching transformers [18, 11, 10], also follows this convention to speed up training. Although training diffusion models directly on raw pixels is technically feasible [19, 20], the preference for latent space modeling stems from practical challenges: raw pixel data often contains high-frequency details and perceptually complex patterns that are computationally intensive and difficult for diffusion processes to model effectively.

**Latent space for diffusion models.**    The compact latent space is crucial for diffusion models to achieve high-quality image synthesis. Latent diffusion models [1] propose to train an autoencoder to map raw pixels to a latent space first, where the latent space is typically $8\times$ spatially downsampled and comes with $4$ channels, reaching a compression rate of $48$. The follow up work on autoencoders, including [5, 21, 22], mainly investigate the channel number and shows that increasing channel number can improve the quality of diffusion samples via larger transformer models. Recently proposed deep compression autoencoders (DC-AE) [23] compress the latent space at more aggressive spatial downsampling rates, e.g., $32$ or $64$, further reducing the training cost of diffusion models.

However, there is a lack of exploration and discussion on the structure of latents for diffusion models. Most autoencoders for diffusion models encode pixels into dense 2D grid-based latents and ignore the underlying non-uniform and sparse structures in natural images, where a background region in an image might be worth less latents than foregrounds. Here in this paper, we study this sparsity as well as visual non-uniformity explicitly with region of interests (RoIs) along with latents for diffusion models, following the sparse visual generation research line [8].

# 3 Methods

Our image synthesis pipeline follows the common practice of latent diffusion models: an autoencoder first that compresses images into a set of latents and RoIs, followed by a generative model that takes noised latents and RoIs as input and predicts diffusion targets. We first describe the design of our *sparse flow autoencoders*, and then introduce *sparse flow-based generative transformers* for modeling joint flow.

## 3.1 Sparse Flow Autoencoders

Given an image $\mathbf{I} \in \mathbb{R}^{H \times W \times 3}$, conventional autoencoders encode it into grid-based latent representations and decode latents back into pixels by the encoder $\mathcal{E}$ and decoder $\mathcal{D}$:

$$\mathbf{z} = \mathcal{E}(\mathbf{I}) \in \mathbb{R}^{H/f \times W/f \times d}, \tag{1}$$

$$\hat{\mathbf{I}} = \mathcal{D}(\mathbf{z}) \in \mathbb{R}^{H \times W \times 3}, \tag{2}$$

where $f$ is the downsampling factor, typically $8$ in practice, and $d$ is the latent dimension. The training for $\mathcal{E}$ and $\mathcal{D}$ is done by minimizing the reconstruction loss $\ell_{\text{rec}}(\mathbf{I}, \hat{\mathbf{I}})$. Variational autoencoders also impose a Kulback-Leibler divergence loss $\ell_{\text{KL}}$ on $\mathbf{z}$ to regularize the latent distribution [24].

**Latent and RoI representation.**  Different from grid-based latent representations, we propose sparse flow variational autoencoders, SF-VAE, to use a sparse set of latents $\mathbf{z} \in \mathbb{R}^{N \times d}$ with their corresponding regions of interests (RoIs) $\mathbf{r} \in \mathbb{R}^{N \times 4}$ to represent an image. The latent space of SF-VAE is simply structured as one flattened dimension space, and the spatial property of latents is characterized in RoIs. The RoIs are represented as bounding boxes in the format of $(x, y, h, w)$, where $(x, y)$ and $(h, w)$ are center points and height and width. The encoder $\mathcal{E}$ now outputs both latents and RoIs, and the decoder $\mathcal{D}$ takes both latents and RoIs as input to reconstruct raw pixels:

$$(\mathbf{z}, \mathbf{r}) = \mathcal{E}(\mathbf{I}), \qquad \hat{\mathbf{I}} = \mathcal{D}(\mathbf{z}, \mathbf{r}). \tag{3}$$

By eliminating the grid-based spatial prior, the number of latents $N$ can be decoupled from image pixels $H \times W$ and can be arbitrarily chosen and further greatly reduced according to our experiments.

**Encoding pixels into latents and RoIs.**  Typical object detectors [25, 26] can encode image pixels into latent representations and RoIs. However, they often lack emphasis on backgrounds and need bounding box annotations as individual supervision. Here, we resort to the SparseFormer architecture [27] to build our encoder $\mathcal{E}$ from scratch, which can encode image pixels into latents and RoIs in *an end-to-end manner without bounding box supervision*. SparseFormer takes early image features $\tilde{\mathbf{I}}$ as input and gradually refines latents $\mathbf{z}$ and RoIs $\mathbf{r}$ via local image features within RoIs $\mathbf{r}$ by several SparseFormer transformer layers, where refinement on $\mathbf{z}$ and $\mathbf{r}$ are both differentiable:

$$(\mathbf{z}^t, \mathbf{r}^t) = \text{SPARSEFORMERLAYER}_t(\tilde{\mathbf{I}}, \mathbf{z}^{t-1}, \mathbf{r}^{t-1}), \tag{4}$$

where $\mathbf{z}^0$ and $\mathbf{r}^0$ are parameters of the model. The RoIs $\mathbf{r}$ are updated using delta-formulation [25] on $(x, y, h, w)$.

**Decoding latents and RoIs back into pixels.**  To learn the distribution of latents and RoIs given an image by just pixel reconstruction, we need to decode them back into raw pixels in *a fully differentiable way*, that is, the reconstruction loss needs to be both differentiable to latents and RoIs. Unfortunately, to our best knowledge, mapping latents and RoIs back into raw pixels in a differentiable way has not been deeply investigated in previous works. We have first tried a simple idea, RoI-aware cross attention, but they struggle to recover high frequency details even in a long training schedule, and cannot learn compact latent RoIs with pixel reconstruction.

Inspired by advance in neural rendering [28, 29], we design our decoder $\mathcal{D}$ with the neural field approach and divide-and-conquer strategy. Specifically, we consider a latent and its RoI indexed by $i$ as a neural field function $\mathcal{F}_{(\mathbf{z}_i, \mathbf{r}_i)} : \mathbb{R}^2 \to \mathbb{R}^3$ whose input is the pixel coordinate and output is the RGB tuple. In other words, a latent and its RoI can be decoded into a pixel image individually. Considering rich high frequency details in natural images [30], we design a neural field based on cosine transform bases similar to 2D discrete cosine transform (DCT) [31] but in a continuous

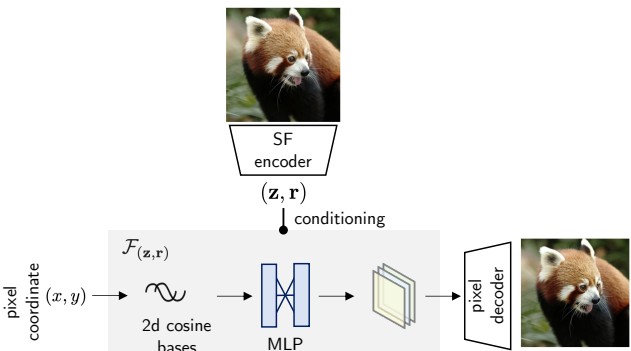

Figure 2: **Structure of our SF-VAE and neural field-based decoder**. For a latent tuple $(\mathbf{z}_i, \mathbf{r}_i)$, the neural field function $\mathcal{F}_{(\mathbf{z}_i, \mathbf{r}_i)}$ first transforms pixel coordinates into 2D cosine bases conditioned on $\mathbf{r}_i$ and then feed these bases into an MLP whose parameters are conditioned on $\mathbf{z}_i$ to get downsampled intermediate features. The pixel decoder then decode these features into raw image pixels. For clarity, we show only a single latent case.

coordinate space. Given a raw pixel's coordinate $(x, y)$ in the image, we first compute relative coordinate $(x', y')$ with regard to the RoI $\mathbf{r}_i$ for a single pixel output:

$$x' = (x - x_i)/w_i, \qquad y' = (y - y_i)/h_i, \tag{5}$$

then compute our customized cosine transform bases $\mathbf{X} \in \mathbb{R}^C$:

$$\mathbf{X}_c = \cos\left[\mathbf{f}_y(c)(y' + 0.5)\pi\right] \cos\left[\mathbf{f}_x(c)(x' + 0.5)\pi\right], \tag{6}$$

where the number of bases $C$ needs to be a squared number, and $\mathbf{f}_y(c)$ and $\mathbf{f}_x(c)$ are the frequency of the $c$-th channel in $y$ and $x$ directions, linearly increasing from 0 to $\sqrt{C} - 1$.

We use a two-layered MLP to transform these cosine bases into final output values, where MLP's weights are conditioned by the latent $\mathbf{z}_i$ through a shared feed forward layer. Note that all these computation can be done with matrix multiplication efficiently. However, we find that directly recovering raw RGB pixels from $\mathcal{F}_{(\mathbf{z}_i, \mathbf{r}_i)}$ is not memory friendly despite being efficient, since we need to "trace" 65536 pixels with their bases and MLP activations for a single latent in a $256 \times 256$ image, which is unrealistic. Therefore, we retarget the neural field to output a downsampled feature map $\mathbf{I}' \in \mathbb{R}^{H/8 \times W/8 \times C}$, use the softmax function to blend different feature maps produced by different latents, and use a upsampling decoder, $\mathcal{D}_{\text{pix}}$, to upsample $\mathbf{I}'$ to the final image $\hat{\mathbf{I}}$ of the input size.

**Overall autoencoder architecture.** In the resulting SF-VAE, a latent does not need to correspond to a fixed spatial region across different images, and we can decouple the number of latents $N$ from the image size and further reduce $N$. Our default number of latents and RoIs is $N = 64$ for a $256 \times 256$ image, where the latent dimension $d$ is 32, reaching a high $96\times$ compression rate[2]. We train SF-VAE in an end-to-end manner using regular VAE loss with a minor difference: we only apply the KL loss to latent variables $\mathbf{z}$ since RoIs $\mathbf{r}$ do not necessarily follow a standard Gaussian distribution. We also design our SF-VAE to be as parameter lightweight as possible. Visualizations in Figure 3 show that SF-VAE can learn semantic and compact latent RoIs and achieve high reconstruction quality, allowing latents to focus on intricate foreground objects.

### 3.2 Sparse Flow Transformers

Now we have compact latent space defined by latent variables $\mathbf{z}$ and RoIs $\mathbf{r}$ for an image, we describe study *how to generate them jointly*. Here we first describe the flow matching framework [9, 32] for continuous data modeling, and then delve into our flow formulation and the design of our flow-based generative transformers.

**Flow matching.** Assume we want to model a target data distribution $p(\mathbf{x})$, flow matching formulates a process starting from a sample drawn from the starting distribution, typically $\mathbf{x}_0 \sim \mathcal{N}(0, 1)$, to a

---

[2]Compression rate is $\sim 85$ if a RoI also counted four values.

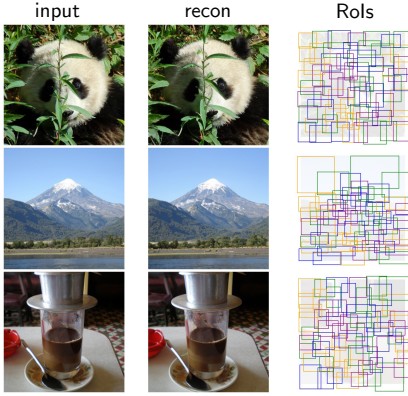
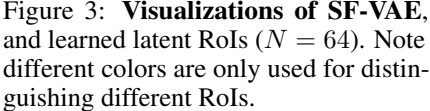
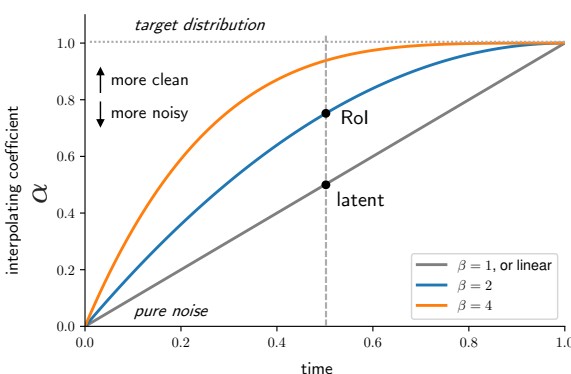

**Figure 3: Visualizations of SF-VAE,** and learned latent RoIs ($N = 64$). Note different colors are only used for distinguishing different RoIs.

**Figure 4: Asynchronous interpolating schedule** for latents and RoIs. RoIs approach target RoIs faster than latents as time $t$ increases using $\beta = 2$ polynomial schedule.

sample from the target distribution $\mathbf{x}_1 \sim p(\mathbf{x})$ by continuous time process:

$$\mathbf{x}_t = \sigma_t \mathbf{x}_0 + \alpha_t \mathbf{x}_1, \tag{7}$$

where $t \in [0, 1]$, $\sigma_t$ and $\alpha_t$ are interpolating coefficients characterizing an interpolating flow. The flow matching requires $\sigma_0 = 1$, $\alpha_0 = 0$, $\sigma_1 = 0$, and $\alpha_1 = 1$. A popular choice is the linear interpolation, $\sigma_t = 1 - t$ and $\alpha_t = t$. The velocity, or the derivation on $\mathbf{x}_t$ in this process is

$$\mathbf{v}_t = \frac{d\mathbf{x}_t}{dt} = \dot{\sigma}_t \mathbf{x}_0 + \dot{\alpha}_t \mathbf{x}_1. \tag{8}$$

The model $\mathcal{F}$ is trained by minimizing the $L_2$ loss between the predicted velocity and the true velocity over sampled points and sampled paths:

$$\mathbb{E}_{t, \mathbf{x}_0, \mathbf{x}_1} \|\mathcal{F}(t, \mathbf{x}_t) - \mathbf{v}_t\|^2. \tag{9}$$

During inference, flow matching draws an initial sample $\mathbf{x}_0 \sim \mathcal{N}(0, 1)$ and solve the oridinary differential equation (ODE) to get the target sample $\mathbf{x}_1$:

$$\frac{d\mathbf{x}_t}{dt} = \mathcal{F}(t, \mathbf{x}_t). \tag{10}$$

Flow matching eliminates the noise introduction in the sampling stage [15] and can model most arbitrary distribution, which is suitable for our RoI distribution modeling. Recent SiT models [10] also shows flow matching as the prediction target with transformers [33] can surpass ones with the diffusion target.

**Joint latent and RoI flow.** Different from SiT models only to model grid-based latents, we need to model both the joint distribution of latents $\mathbf{z}$ with their RoIs $\mathbf{r}$ by SF-VAE in our flow-based generative transformers. The latents $\mathbf{z}$ in SF-VAE are not structured as a grid, and their positional properties are encoded in RoIs $\mathbf{r}$. Here, we keep most of the transformer architecture in SiT unchanged but retarget it to take both latents and RoIs as input and predict the velocity of both them:

$$(\hat{\mathbf{v}}_{z,t}, \hat{\mathbf{v}}_{r,t}) = \mathcal{F}(t, \mathbf{z}_t, \mathbf{r}_t), \tag{11}$$

where $\mathbf{z}_t$ and $\mathbf{r}_t$ are interploated latents and RoIs following Equ 7 at time $t$, $\hat{\mathbf{v}}_{z,t}$ and $\hat{\mathbf{v}}_{r,t}$ are estimated velocity of latents and RoIs. We name it as SF-SiT for modeling joint latent and RoI flow. We choose the initial distribution of latents and RoIs both to be standard Gaussian distributions for simplicity.

Given the number of latents $N$ already being highly reduced (e.g., 64), we do not perform any latent "grouping" and "ungrouping" operations in SiT to reduce computation. In other words, a latent from SF-VAE directly corresponds to a token in the transformer in SF-SiT.

**RoI-based positional encoding.** Since we get rid of the grid-based latents, it is crucial to inject positional information dependent on RoIs $\mathbf{r}_t$ into the transformer. Otherwise, the latents $\mathbf{z}$ becomes fully permutation invariant due to the transformer's architecture. We use the sinusoidal positional encoding on four values of RoIs, $(x, y, h, w)$, and concatenate them together. We add them as position encoding to a corresponding token in the transformer. It is worth noting that this positional encoding is totally based on RoIs and therefore changes for the same latent across different training and inference timesteps as RoIs move across different timesteps.

**Asynchronous interpolation schedule.** The time $t$ indicates the signal-to-noise ratio (SNR) in the interpolation schedule, where $t = 0$ means pure noise and $t = 1$ means target data. Recall that we need to model both $\mathbf{z}$ and $\mathbf{r}$ and therefore SF-SiT are required to handle pure noise RoIs at time $t = 0$ as well as not-so-clean RoIs in early $t$ period. This differs from the regular SiT where the position information is always clean, and the model have strong position information given at any $t$ to predict the latent velocity to denoise them. Therefore, we design an asynchronous interpolation schedule for RoIs compared with latents, where we interpolate RoIs to approach the target RoIs faster than latents in a polynomial way:

$$\sigma_{\mathbf{r},t} = (1 - t)^{\beta}, \qquad \alpha_{\mathbf{r},t} = 1 - (1 - t)^{\beta}, \tag{12}$$

where $\beta$ is a control parameter, and reduces to the synchronous linear interpolation when $\beta = 1$. We keep the linear interpolation schedule untouched for latents:

$$\sigma_{\mathbf{z},t} = 1 - t, \qquad \alpha_{\mathbf{z},t} = t. \tag{13}$$

This asynchronous interpolation schedule for latents and RoIs allows the velocity prediction for $\mathbf{z}_t$ to be more position-aware with cleaner RoIs as $\mathbf{r}_t$ approaches the target RoIs faster, as shown in the Fig 4. It is worth noting that the asynchronous interpolation schedule only affects the training phase, i.e., the velocity computation of latents and RoIs. The inference logic is the same as regular SiT, where we solve the ODE to sample the target latents and RoIs. To force the model to learn the accurate RoI distribution, we also impose an additional $L_1$ between the predicted RoI velocity and the ground-truth, with a balancing weight $w_{L_1}$, as proposed by LayoutFlow [34] to model bounding box flow more accurately in layout generation.

## 4 Experiments

To verify the feasibility of synthesizing images with sparse non-grid latents, we conduct experiments on the standard ImageNet benchmark [35], mainly on $256 \times 256$ images. We first discuss our autoencoders, SF-VAE, and then present the results of our generative model, SF-SiT. Note that our aim is not to achieve new state-of-the-art results on a benchmark, but to explore sparsity in image generation pipeline.

### 4.1 Sparse Flow Autoencoders

**Setup.** As discussed in previous section, our SF-VAE consists up of a SparseFormer encoder $\mathcal{E}$ and a neural field-based decoder $\mathcal{D}$, where the latent space defined by latent variables and RoIs. To keep as lightweight as possible, we design a SparseFormer encoder of 8 blocks with transformer dimension 512 to extract latents and RoIs, where the leading 4 blocks extract RoI regional features from the image. The neural field decoder also consists of 8 transformer blocks of 512 dim, and then

Table 1: **Reconstruction results** on 50K ImageNet validation samples.

| resolution | method | latent shape | params | rFID↓ | PSNR↑ | SSIM↑ | LPIPS↓ |
|---|---|---|---|---|---|---|---|
| $256 \times 256$ | SD-VAE-ema-f8 | $32 \times 32 \times 4$ | 84M | 0.63 | 24.98 | 0.804 | 0.062 |
| | DC-AE-in-f32c32 | $8 \times 8 \times 32$ | 323M | 0.77 | 23.92 | 0.765 | 0.086 |
| | DC-AE-mix-f32c32 | $8 \times 8 \times 32$ | 323M | 0.96 | 23.75 | 0.763 | 0.088 |
| | **SF-VAE** | $64 \times 32$ | 133M | 0.70 | 23.24 | 0.743 | 0.085 |
| $512 \times 512$ | SD-VAE-ema-f8 | $64 \times 64 \times 4$ | 84M | 0.19 | 27.36 | 0.849 | 0.061 |
| | DC-AE-in-f32c32 | $16 \times 16 \times 32$ | 323M | 0.20 | 26.23 | 0.815 | 0.078 |
| | **SF-VAE** | $256 \times 32$ | 133M | 0.29 | 25.03 | 0.787 | 0.088 |

produce dynamic MLP's parameters, where the hidden dimension of two-layer MLP is 64. The output of the neural field decoder is then decoded into raw pixels by $\mathcal{D}_{\text{pixel}}$ of conventional VAE decoder architectures [1] but we reduce the dimension of most convolutional layers. The computational cost of the resulting SF-VAE decoder is 153GFLOPs for a $256 \times 256$ image, where the SD-VAE-ema-f8 decoder needs 311GFLOPs. The latent $\mathbf{z}$ is 32-dim, and the default number of latents is 64. The loss follows the convention of VAE in latent diffusion models, with the $L_2$ loss for pixel reconstruction, perceptual loss [36] and GAN adversarial loss [37] for visual perceptual details, as well as KL divergence [24] for the latent regularization on $\mathbf{z}$. We train the SF-VAE on the ImageNet training set with a batch size of 128 for 320K iterations (equivalent to 32 epochs) with a learning rate of $10^{-4}$. We perform reconstruction FID evaluation [38] on the ImageNet validation 50K samples.

**Reconstruction results.** We compare our SF-VAE with the popular SD-VAE-f8 [1] that comes with a downsampling rate of 8, resulting in a latent shape of $32 \times 32 \times 4$. We also include the recently proposed deep compression autoencoder (DC-AE) [23] with a latent shape of $8 \times 8 \times 32$ for comparison, which is exactly the same as our default latent shape. The results are shown in Table 1.

The reconstruction FID of our SF-VAE is 0.70, which is competitive with the SD-VAE (rFID 0.63) with 1024 latents. Also, if counting the latent dimension together, the SF-VAE comes with $64 \times 32 = 2048$ units, much less than SD-VAE's $32 \times 32 \times 4 = 4096$ units. Compared with DC-AE also trained on ImageNet samples and with the same latent shape, our SF-VAE achieves a lower rFID score while maintaining significantly fewer parameters.

More importantly, the reconstruction results further demonstrate that non-grid latents are also effective for image reconstruction. These findings align with recent studies, which show that explicit 2D structured latents are not necessary for high-quality reconstructions and token numbers can be extremely compressed [39–41]. However, our methods does not necessarily belong to the 1D tokenization family, since there are explicit geometric properties, i.e., RoIs, associated with each latent. These 2D RoI representations also maintain the translation equivariance property of this pipeline, while 1D tokenization methods typically fail to address this point.

**Varying number of latents and higher image resolution.** As discussed before, SF-VAE enjoys the flexibility of decoupling the number of latents from a specified image resolution. Here, we investigate our SF-VAE with more aggressive latent reduction for a fixed resolution of $256 \times 256$ in Table 2. We do not change the architecture of our SF-VAE, but only changes the number of latents from 64 to 48 and 32 and maintains the latent dimension unchanged. To reduce training costs, we initialize SF-VAE with new latent configurations from pre-trained weights of the 64-latents model, and finetune them for 20K iterations. Here, we find 32 tokens have already been able to reconstruct pixels with an rFID of 1.70, and 48 tokens give a competitive 0.99 rFID.

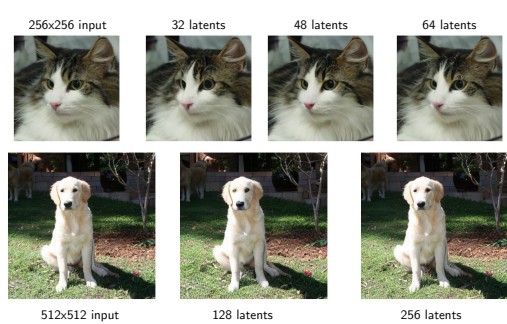

Figure 5: SF-VAE reconstructions with different latent numbers. Better zoom in.

Table 2: Token numbers for SF-VAE reconstructions.

| resolution | #latent | rFID |
|---|---|---|
| | 128 | 0.35 |
| $256 \times 256$ | 64 | 0.70 |
| | 48 | 0.99 |
| | 32 | 1.70 |
| $512 \times 512$ | 256 | 0.66 |
| | 128 | 1.29 |

Table 3: Effects of asynchronous interpolating schedule.

| interpolating schedule | FID↓ |
|---|---|
| linear, $\beta = 1$ | 44.4 |
| async, $\beta = 2$ | **42.0** |
| async, $\beta = 4$ | 44.6 |
| async, $\min(2 \times t, 1)$ | 54.0 |
| async, $\min(4 \times t, 1)$ | 55.9 |
| decoupled, RoIs first | 63.3 |

Table 4: Effect of $L_1$ loss.

| $w_{L_1}$ | FID |
|---|---|
| 0.0 | 43.9 |
| 0.2 | **42.0** |
| 0.5 | 42.6 |
| 1.0 | 45.1 |

Table 5: Comparison with diffusion/flow-based transformers with grid-based latents under 400K training iterations without CFG. *SiT reproduced using official code. † sampled with ODE Euler sampler with NFE=250 for fair comparison. Flops are calculated with one single NFE.

| method | #tokens | #params | flops (G) | FID↓ | IS↑ |
|--------|---------|---------|-----------|------|-----|
| DiT-B/2 | 256 | 130M | 21.8 | 43.5 | - |
| SiT-B/2 | 256 | 130M | 21.8 | 33.0 | - |
| SiT-B/2*† | 256 | 130M | 21.8 | 37.0 | 39.6 |
| **SF-SiT-B**† | 64 | 138M | 5.9 | **33.6** | **41.4** |
| DiT-XL/2 | 256 | 675M | 114.4 | 19.5 | - |
| SiT-XL/2 | 256 | 675M | 114.4 | 17.2 | - |
| SiT-XL/2*† | 256 | 675M | 114.4 | 18.8 | 70.9 |
| **SF-SiT-XL**† | 64 | 685M | 29.3 | **18.6** | **71.6** |

We also study SF-VAE on higher image resolution of $512 \times 512$ in Table 2. We first scale up the number of latents to 256 similar to dense grid-based VAEs, where latent numbers are increasing quadratically regarding input image size. Then we reduce the number of latents to 128. We find that our SF-VAE can still achieve competitive reconstructions for higher images in Figure 5.

## 4.2 Sparse Flow Transformers

**Setup.** Our SF-SiT generally follows the design of original SiT transformers [10], where the input and output are retargeted to latents $\mathbf{z}$ and RoIs $\mathbf{r}$, and the velocity of them, $\hat{\mathbf{v}}_z$ and $\hat{\mathbf{v}}_r$. We mainly experiment SF-SiT-B and SF-SiT-XL variants, whose configurations strictly follow the original SiT-B and SiT-XL. Since SF-SiT are now required to output RoI velocity besides latent velocity, we add an additional AdaLN head [42] to predict RoI velocity, where the output is also initialized to zeros similar to latent prediction.

We train SF-SiT variants from scratch on ImageNet training set with 64 latents and RoIs from SF-VAE, unless otherwise specified. In SF-SiT, each latent directly corresponds to a token in the transformer blocks, as our architecture eliminates extra patchifying operations. The training configurations are the same as the original SiT models, i.e., the global batch is 256 and the Adam optimizer [43] with constant learning rate $10^{-4}$. We set the default $\beta$ to 2 in the asynchronous interpolating schedule and the weight $w_{L_1}$ to 0.2 for the additional L1 loss on RoI velocity prediction. We evaluate our SF-SiT using ADM evaluation pipeline [44] and report FID with 50K samples.

**Comparison with grid-based diffusion/flow transformers.** We first compare our methods under the 400K iteration budget with grid-based DiT/SiT baseline [11, 10] in Table 5 without classifier-free guidance (CFG). It is worth noting that original SiT models adopt the SDE sampler during inference and report their results. This differs from SF-SiT models, which use simple ODE sampler for latent and RoI inference. Due to the lack of publicly available intermediate SiT checkpoints, we reproduce SiT-B and SiT-XL variants using the official code, and inference samples with ODE Euler sampler. We also align the number of function evaluations (NFE) in SF-SiT and SiT models to 250 to make fair comparison.

As shown in Table 5, SF-SiT models, despite operating on significantly reduced latents, achieve performance on par with SiT models using the Euler ODE sampler. We experimented SF-SiT models with the original SiT's SDE sampler but observed degraded RoI quality and visual distortion in decoded images. We hypothesize this sensitivity arises from that SF-VAE encoded RoIs being deviating from standard Gaussian distributions, making them incompatible with the noise introduced by SDE sampling. Overall, these findings suggest that jointly modeling the flow of latents and RoIs have competitive performance paired with ODE-based samplers.

**Ablation studies.** For all ablation studies, we resort to SF-SiT-B variant with 200K budget due to the limited computational resource. We also inference samples in ablations with the Heun ODE solver with 50 NFE for efficiency.

**Asynchronous interpolation schedule.** Our proposed SF-SiT introduces additional RoIs $\mathbf{r}$ as additional modeling targets. As discussed before, we propose the asynchronous interpolation schedule

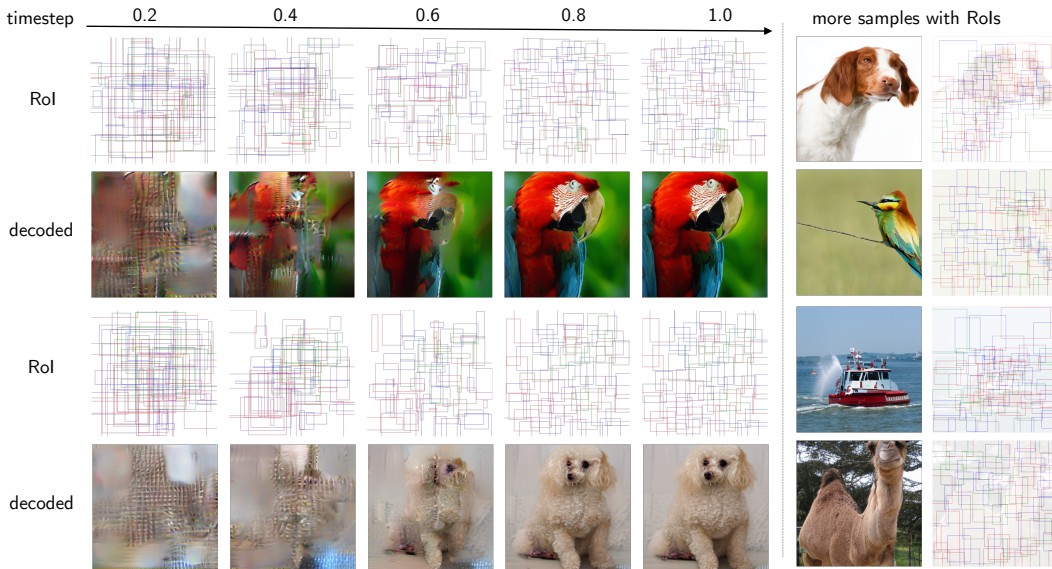

Figure 6: Visualization of RoI flows and decoded images with SF-SiT-XL and CFG=4.0.

for $\mathbf{r}$ to be more compatible in the joint modeling with the latents $\mathbf{z}$. Here we investigate the effect of the asynchronous interpolation schedule in Table 3. We can find that the asynchronous interpolation schedule $\beta = 2$ for $\mathbf{r}$ gives the best result. In contrast, synchronous interpolation ($\beta = 1$) between $\mathbf{r}$ and $\mathbf{z}$, or purely linear interpolation for both, results in the inferior FID. We also try more aggressive asynchronous schedule, where the interpolated RoI $\mathbf{r}$ reaches the target RoI linearly but at the earlier timestep, e.g., $t = 0.25$ for $\min(4 \times t, 1)$. These clamped linear schedules are also asynchronous, where we suppose to sample RoIs first and then sample latents in a hard way, leading to significant degradation in FID. Finally, the decoupled interpolation schedule indicates that we first schedule RoI denoising separately and individually when $t$ is in $[0, 0.5]$ and then schedule latent denoising when $t$ is in $[0.5, 1]$, where the denoising processes for $\mathbf{r}$ and $\mathbf{z}$ are fully decoupled and disjoint, leading to the worst FID. These findings show that the joint flow of latents and RoIs is crucial to good performance in SF-SiT framework but also needs soft asynchronous interpolation between $\mathbf{z}$ and $\mathbf{r}$ to have better performance.

**RoI velocity $L_1$ loss.** We also investigate the effect of $L_1$ loss on the RoI velocity prediction, as proposed by. In line with [34], the $L_1$ loss weight 0.2 gives the best result.

**Longer training schedule.** We further train the largest SF-SiT variant, SF-SiT-XL, with longer training schedule. To accelerate convergence on limited hardware resource, we use REPA [45] for our SF-SiT-XL training. We use DINOv2-B/14 [46] as our REPA align target. Since our SF-SiT

Table 6: Class conditional generation results on $256 \times 256$ ImageNet. $P$ refers to precision and $R$ refers to recall following [44].

| method | #tokens | train steps | FID↓ | IS↑ | P↑ | R↑ |
|---|---|---|---|---|---|---|
| LDM-4 [1] | - | - | 10.56 | 103.5 | 0.71 | 0.62 |
| DiT-XL/2 [11] | 256 | 7M | 9.62 | 121.5 | 0.67 | 0.67 |
| SiT-XL/2 [10] | 256 | 7M | 8.26 | 131.6 | 0.68 | 0.67 |
| SiT-XL/2 w/ REPA [45] | 256 | 1M | 6.4 | - | - | - |
| **SF-SiT-XL** w/ REPA | 64 | 1.4M | 7.35 | 131.9 | 0.70 | 0.66 |
| DiT-XL/2 (CFG=1.5) | 256 | 7M | 2.27 | 278.2 | 0.83 | 0.57 |
| SiT-XL/2 (CFG=1.5) | 256 | 7M | 2.06 | 277.5 | 0.83 | 0.59 |
| **SF-SiT-XL** w/ REPA (CFG=1.5) | 64 | 1.4M | 2.99 | 279.2 | 0.82 | 0.54 |
| **SF-SiT-XL** w/ REPA (CFG=1.375) | 64 | 1.4M | 2.76 | 247.7 | 0.81 | 0.58 |

does not form grid-based feature maps, we need to transform intermediate representations in SF-SiT transformer blocks into a grid using the time-dependent interpolated $\mathbf{r}_t$ as positional properties. Here, we use a simple cross attention from dense queries on grids to SF-SiT tokens with the positional embedding based on $\mathbf{r}_t$ added into keys and values. Dense grid queries are then passed through an MLP to align with DINOv2.

Table 6 shows SF-SiT-XL (w/ 64 latents) results with longer training budget. Note that here we only want to show the potential of SF-SiT-XL on our limited budget with the fast convegence of REPA (it have already took 7 days on 8 A100s to complete 1.4M steps). SF-SiT-XL reaches the competitive performance (7.35 FID w/o CFG and 2.76 w/ CFG) with significantly reduced tokens and computation used. It is expected to have better results when training for more iterations and we will investigate them further when computational resources are available.

## 5 Conclusion

In this paper, we proposed a paradigm for visual generative modeling by using non-grid sparse latents with their positional properties, challenging the dense grid convention for image synthesis. We first designed sparse flow variational autoencoder, SF-VAE, which encodes raw image pixels into latents together with their RoIs. With SF-VAE, we can compress $256 \times 256$ images with down to 32 latents for good reconstruction fidelity. We then propose sparse flow-based generative transformers, SF-SiT, to model the joint flow of latents and RoIs with the highly reduced latent space. With carefully-designed asynchronous interpolation schedule and loss for RoIs, SF-SiT models have competitive performance compared with diffusion/flow transformers. Although not targeting new state-of-the-art results, SF-SiT-XL with just 64 tokens still reaches a competitive 2.76 FID on ImageNet with longer training schedule. We hope that our work can facilitate further research directions on non-grid and sparse generative methods, as well as sparse approaches for generation in other domain, e.g., audio and video.

**Limitations.** Due to the lack of computational resources, we only train our SF-SiT-XL variant with the help of representation alignment (REPA) for fast convergence, and under fewer tokens settings, it might require more training iterations to converge with RoI modeling compared with grid-based SiT/DiTs for large parameter models. Also, the large scale text-to-image experiments remain exploration in the future. As this paper mainly aims for visual generative modeling, it inherits safety risks regarding its generated content. As the standard ImageNet being the training dataset, which is known to be a relatively safe for academic purposes, these concerns might not be fully addressed in our current framework.

**Acknowledgement.** This project is supported by the National Research Foundation, Singapore under its NRFF Award NRF-NRFF13-2021-0008.

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
