# OpenReview forum: "Sparse Image Synthesis via Joint Latent and RoI Flow"
_NeurIPS.cc/2025/Conference — NeurIPS 2025 poster_

### Official Review · Reviewer_Jekt · 2025-06-23

**Clarity:** 4
**Significance:** 4
**Originality:** 4
**Rating:** 5
**Confidence:** 4

**Summary:**

This paper focuses on developing a compact encoding for images that captures their sparse nature. To achieve this, the authors introduce sparse flow autoencoder (SF-VAE) to encode an image using a small number of learnable RoI tokens their bounding boxes, which is in contrast to the typically pixel-aligned grid. Additionally, the author introduce a sparse flow-based transformer (SF-SiT) which achieves competitive performance to recent diffusion/flow-based transformers in terms of image generation while using a more compact representation.

**Questions:**

Please see above.

**Ethical Concerns:**

["NO or VERY MINOR ethics concerns only"]

**Final Justification:**

The authors has provided additional experimental results addressing the other reviewers' requests. The results seem supportive of main claims made by the paper.

**Limitations:**

The authors answered in the checklist that there is discussion on the limitations, but there does not seem to be one included.

**Quality:**

3

**Strengths And Weaknesses:**

## Strength

The general motivation for a more compact representation based on RoI and bounding box makes sense. The reconstruction results seem to support the viability of the RoI encoding approach. Also, the idea of using a neural renderer for making the decoder differentiable w.r.t. the RoI sounds new and interesting. The generation results also demonstrates competitive bench mark results.

## Weakness

There does not seem to be any major weakness for this paper. But there are a few minor questions:

1. The RoIs are overlapping with each other quite a lot except the ones near the boundary. It would be interesting to learn more about what each RoI is decoded to, are they memorizing different structures in the image (like a silhouette of certain objects)?
2. The upsampling approach for decoding the MLP output feature seems to be producing some blobby artifact or at least makes the reconstruction result not reproducing finer details in the input fully (e.g., the black blobs at the river bank and on the glass in Figure 3). Do the authors have any proposal for how to improve this?

---

> ### Author Rebuttal · Authors · 2025-07-31
>
> We would to thank you for your positive comments and review on our submission. We address the reviewers' concerns as following:
>
> **1. It would be interesting to learn more about what each RoI is decoded to, are they memorizing different structures in the image?**
>
> This is a good suggestion. Indeed we had visualized each RoI individually a lot during developing this approach and we find that each RoI corresponds to low-level spatial patterns and often come across boundaries of different objects. Since we do not impose any semantic loss/alignment, this behavior is expected. Our method only compresses perceptual details (spatial patterns/textures, not semantic) into latents with RoIs. The RoI size also varies when coming into areas of different frequencies, e.g., low-frequency pixel area coverred by large RoIs. We will add single-RoI-decoded visualizations and discussion on these to the revised version.
>
> **2. How to improve blobby artifact or reproducing details fully?**
>
> It is a great question but also very hard for every researcher on better VAE. If we run today's advanced VAE such as SD-VAE or Flux-VAE, we can find that these reconstructed results also do not reproduce every details in the input fully. VAE commonly is designed for lossy compression since the latent space size is greatly smaller than the input size. Having said that, we can also increase the model's parameters and computations, especially SFVAE's decoder, to lower the gap. In the submission, we design the pixel decoder to be as light as possible (See response to Q.2. from Reviewer pbLv) and this might be the bottleneck for reconstruction capability. Alternatively, we can increase the latent space size to contain more information, which is beneficial for reconstructing pixels accurately. But this way might hinder the convergency the following diffusion model learning since the sample size will become larger.

---

> > ### Comment · Reviewer_Jekt · 2025-08-03
> > **Keeping my score**
> >
> > Thank you for the response. I've also read the other reviews, and I would keep my rating.

---

### Official Review · Reviewer_BxLE · 2025-06-24

**Clarity:** 3
**Significance:** 3
**Originality:** 3
**Rating:** 4
**Confidence:** 3

**Summary:**

The paper proposes to learn sparse non-grid latent representation for image synthesis. A sparse autoencoder learns a set of non-grid latents and RoIs via reconstruction loss, where the decoder is designed with the neural field approach. A flow-based transformer models the joint flow of the latents and RoIs in the denoising process.

**Questions:**

- The Table 2 ablates the impact of reducing the number of latents from 64 to 32 for resolution 256 x 256. What if the number is increased to 128? It would be interesting to include the #latent=128 result, as it would match the latent units of SD-VAE and allow for a more direct and fair comparison.

**Ethical Concerns:**

["NO or VERY MINOR ethics concerns only"]

**Final Justification:**

After considering the rebuttal, I would like to maintain my original score. The additional experimental results show that the method works with more tokens for reconstruction. However, it remains unclear whether it performs well for higher-resolution generation, as results for 512×512 are missing.

**Paper Formatting Concerns:**

No formatting concern

**Quality:**

3

**Strengths And Weaknesses:**

**Strengths**
- The paper is well-motivated. Many real-world images often show non-uniform information density so it is natural to explore new methods with non-grid representation.
- The exploration of the sparse non-grid latent representations for image synthesis  could make a meaningful contribution to the research community. Ideas such as decoding latents and RoIs in a neural rendering manner and the asynchronous interpolating schedule for RoIs both offer new insights.
- The paper is well-written with clearly presented figures. The idea flow is also very clear and easy to follow.

**Weaknesses**
- For results reported in Table 1, there is no explanation of the difference between *DC-AE-in-f32c32* and *DC-AE-mix-f32c32*. And the numbers for DC-AE on ImageNet differ slightly from the numbers presented in the DC-AE paper.
- The Figure 5 could be more informative. For example, it would be interesting to compare the visualization of the RoIs w.r.t to different latent numbers, along with the corresponding qualitative metrics (e.g., PSNR).
- The description of the experimental setting of SF-SiT could be clearer. The resolution info is missing for the results reported in Table 5. Since the SF-SiT-XL in Table 6 is also trained with REPA, it could be clearly indicated in the table. Also, it would be good to include the 512 x 512 results to demonstrate the model's high-resolution generative capability.

---

> ### Author Rebuttal · Authors · 2025-07-31
>
> We would like to thank reviewer's time for reviewing and comments on our work. We would like to address concerns:
>
> **Q.1. The difference between DC-AE-in-f32c32 and DC-AE-mix-f32c32 and DC-AE results differs from the original paper.**
>
> The difference between DC-AE-in-f32c32 and DC-AE-mix-f32c32 mainly comes from the training dataset. The DC-AE-in-f32c32 is trained on ImageNet training set and DC-AE-mix-f32c32 is trained on their own mixed dataset according to DC-AE paper. Our reported DC-AE results differs from the original paper because we report different models on standard 50K ImageNet **validation** set while DC-AE original paper evaluates models on ImageNet **test** set (from publicly-available code), which is an uncommon practice.
>
> **Q.2. The Figure 5 could be more informative.**
>
> Thank you for your advice on Figure 5 and we will revise it to be more informative.
>
> **Q.3. The experimental setting of SF-SiT and Table 6 could be more clearer. It would be good to include 512x512 results**
>
> Thank you for your pointing these out and valuable suggestions! Results in Table 5 are on 256x256 resolution. We will add REPA to SF-SiT-XL in Table 6 to clearly indicate it is trained with REPA. If resources available in nearly future, we will finetune SF-SiT-XL on 512x512 resolutions from SF-SiT-XL w/REPA pre-trained on 256x256 with moderately increased latent tokens and compare with other methods.
>
> **Q.4. What if the token number in Table 2 is increased to 128?**
>
> We finetune SF-VAE tokens number 128 variant for 20K iterations on ImageNet 256x256 per your valuable advice and report results here:
>
> | method                 | rFID | PSNR  | SSIM  | LPIPS |
> |------------------------|------|-------|-------|-------|
> | SD-VAE-ema-f8          | 0.63 | **24.98** | **0.804** | 0.062 |
> | SF-VAE, 64 tokens      | 0.70 | 23.24 | 0.743 | 0.085 |
> | **SF-VAE, 128 tokens** | **0.35** | 24.81 | 0.798 | **0.060** |
>
> We can see that SF-VAE w/ 128 tokens achieves competitive performance, especially rFID, compared with SD-VAE-ema-f8 on the same compression rate.

---

> > ### Comment · Reviewer_BxLE · 2025-08-04
> >
> > Thank you for the response and the additional experiment results. I agree the results demonstrate that SF-VAE-128 achieves competitive performance with SD-VAE when the compression rate matches.  I will maintain my original score.

---

### Official Review · Reviewer_zU5o · 2025-06-25

**Clarity:** 3
**Significance:** 3
**Originality:** 3
**Rating:** 5
**Confidence:** 4

**Summary:**

This paper discusses image synthesis from the perspective of sparsity. Specifically, the latent and position attributes of the image are modeled separately, and flow matching is used to generate the latent and position respectively. Compared with the previous grid-based modeling, it has lower computing resources and competitive performance.

**Questions:**

See weaknesses, more analysis will help to better understand how it works

**Ethical Concerns:**

["NO or VERY MINOR ethics concerns only"]

**Final Justification:**

This paper is overall valuable, with a novel idea, although some experiments may be somewhat incomplete.

**Limitations:**

yes

**Quality:**

3

**Strengths And Weaknesses:**

Strengths:
1. Modeling images as a combination of latents and RoIs from an unstructured view is insightful. This approach could inspire future multimodal learning (e.g., aligning latents with semantics) or efficient video generation, making it valuable to the community.

2. The use of neural rendering to reconstruct images from latents and RoIs is creative. The model achieves 96× compression while maintaining performance close to state-of-the-art VAEs. The proposed RoI-based positional encoding and scheduler are also well-designed, enabling performance comparable to the SiT baseline using only 64 tokens (vs. 256).

3. The paper is clearly written and includes thorough ablations on latent size, resolution, and scheduling.

Weaknesses:
1. Due to space limits and no supplementary material, many implementation details are missing. For example, as author provides RoI visualization in Figure 3, it's unclear what each RoI-latent pair represents—do they correspond to structured objects or low-level features? Some analysis would help clarify this.

2. Some recent work in discrete image generation explores compressing token representations (e.g., 1D tokenizers [1]). It would be interesting to hear the authors’ views on how their method compares to these approaches.

3. Looking forward to open source and further work.

Reference:

[1] An Image is Worth 32 Tokens for Reconstruction and Generation

---

> ### Author Rebuttal · Authors · 2025-07-31
>
> Thank you for your comments and advice on our submission. We will add more implementation details as much as possible in future version per your advice to make the paper more clear and less confusing. We will open source our work  soon upon acceptance. We  address your concerns as following:
>
> **Q.1. Do RoIs correspond to structured objects or low-level features?**
>
> As for now, RoIs in SF-VAE mainly corresponding to low-level features (such as textures and edges, mostly frequency based). From Fig 3, we can see RoIs are mainly influenced by the pixel frequency within their covered regions. Generally, RoIs become dense towards high frequency areas while a single RoI becomes large  when covering low frequency content. But overall, RoIs here do not have many strong semantics in RoIs since we do not impose any semantic alignment loss.  We will provide more RoI visualizations and analysis in future version.
>
> **Q.2. How our method compare with discrete image compressing representations**
>
> Thank you for rising an insightful discussion here. First of all, our methods (including SF-VAE and SF-SiT) are all continuous-based and designed for diffusion while TiTok and other 1D image representations are discrete tokens are designed for MaskGiT or autoregressive modeling, which is significantly different. These two method are not directly comparative since the space of discrete representations is much smaller than the continuous one. So we'll see a large margin between SF-VAE and TiTok on all token number settings (see response to Q.1. from Reviewer pbLv) and this is expected.
>
> Despite the high-level idea that TiTok and SF-VAE share, they are exploring two different pathways to compress image pixels. TiTok uses transformers to learn compact tokens, which is highly data-driven and lose spatial structures since images are reformatted to 1D representations. SF-VAE exploits latent-and-RoI pairs to learn continuous representations (RoIs are also continous) and maintains the translation equivariance property, which we believe it is suitable and beneficial for image modeling. We will add more discussion on this point

---

> > ### Comment · Reviewer_zU5o · 2025-08-04
> >
> > I have carefully read the author's responses to my questions and the rebuttal to other reviewers. My concerns are addressed and I will maintain my score.

---

### Official Review · Reviewer_pbLv · 2025-06-30

**Clarity:** 3
**Significance:** 2
**Originality:** 2
**Rating:** 3
**Confidence:** 4

**Summary:**

This paper proposes to learn a latent generative model for images where, in contrast to most previous work, the latents are not structured in a 2D grid. To learn such model, similarly to other latent space models (e.g., LDM), the paper proposes a two-stage approach: the first stage learns a Sparse Flow VAE (SF-VAE), which encodes an input image into a sparse set of 1D latent codes / tokens and RoI representations; the second stage learns a flow model to sample from the target (joint) distribution of latents and RoIs, by adapting a transformer-based model (SiT) to parameterize the velocity field. The paper refers to the latter as Sparse Flow SiT (SF-SiT).
Explicitly modeling RoIs introduces additional complexity in learning a joint velocity field, especially at timesteps close to t=0, as the model has to rely on a noisy RoI representation to predict a position-aware latent velocity. To alleviate this issue, the paper proposes an asynchronous interpolating schedule, by designing an interpolation scheme where RoIs approach the target distribution more quickly, compared to the corresponding latents.
The paper evaluates the proposed approach on the ImageNet generation benchmark, at two different resolutions (i.e., 256x256 and 512x512), showing competitive results (gFID) when compared with baselines that do employ a grid-structured latent space, while reducing number of latent tokens and training compute.

**Questions:**

1. Can the authors compare the proposed approach against other 1D image tokenization strategies? One straightforward example could be the already mentioned "TiTok" [B], which also evaluates on the ImageNet 256x256 benchmark.
2. Can the authors compute the total number of FLOPs for sampling an image in pixel space? Which means, also including the necessary operations to convert your 1D tokens into 2D feature maps via the neural field decoder into the total amount of flops.
3. Can the authors report (cf. Table 6) the result (gFID) obtained with the SiT-XL baseline using REPA for faster convergence and CFG scale > 0? It will be helpful to really show the potential of the proposed SF-SiT-XL with limited compute.
4. Can the authors provide more insights on why we need to (explicitly) model the distribution of RoIs? This question is also partially motivated by the fact that the RoIs (cf. Figure 6) seem to flow to a fairly even distribution where each token still corresponds to a small patch of your image. Ablating this choice can be a very good starting point.
5. Can the authors provide a more convincing discussion of the issue of safeguarding their image generation approach?

In general, I would consider increasing my score upon seeing a comparison with other previous works on 1D image tokenization, which is currently missing from the paper. Also, the issue of safeguarding is not handled well. Thus, (1) and (5) seem to be the most relevant point to address. Addressing other points (2-4) would be helpful to have more insights on the significance of the proposed approach.

**Ethical Concerns:**

["NO or VERY MINOR ethics concerns only"]

**Final Justification:**

As I already stated in the response to the authors, I am not completely convinced by this paper. I agree with reviewer Rnro that the learned RoIs do not seem to be easily interpretable. The authors are trying to argue that interpretability of the RoIs is not needed for reconstruction or generation, but I do not find this convincing. Even for reconstruction and generation, we do want to know what the role of the main technical innovation of a proposed approach is. In my view, this desire extends beyond applications to scene understanding.
Without any precise answer on the role of the RoIs, I believe that a paper that proposes RoIs as its main technical innovation cannot stand on its own. Even if they model low-level details, we would need some analysis to show this quantitatively and some argumentation why this is important. I thus maintain my reject recommendation, upgrading it slightly to borderline reject in light of the (more) convincing points in the rebuttal.

**Limitations:**

The paper is currently missing a paragraph discussing (or briefly summarizing) the limitations of the approach. Although this is already shortly mentioned in the conclusion, it would be helpful to have a separate paragraph mentioning that the goal of this work is not to target SotA generation results, but rather to demonstrate the applicability and scalability of a new image tokenization approach. Another additional point should make it clear that the initial hypothesis has been verified on limited compute (i.e., training time), but it is still to demonstrate whether this approach can scale to more compute and surpass grid-based latent representations, while being faster and less memory-intensive.

I disagree with the answer to question 11 in the questionaire. This is an image generation model and as such the question of safeguards is indeed a relevant issue. While the safeguards needed here are likely no different than those for most of the models compared to, I do believe that this issue cannot be swept under the rug quite this easily. The paper should at least reflect on whether safeguards may be necessary and share these reflections with the readers (and reviewers). References to relevant work on safeguarding of image generators would also improve the paper.

**Paper Formatting Concerns:**

No major concerns found.

**Quality:**

2

**Strengths And Weaknesses:**

Strengths:
+ The paper is quite well-written and easy to follow. All the introduced architectural components are presented in detail, making the technical contributions of the paper easy to grasp.
+ The idea of encoding an image into a sparse set of latents instead of a lower-resolution latent "image" is definitely interesting and worth exploring. In fact, while a 2D latent space can benefit from the inductive bias of convolutions in U-Nets, as discussed in [A], it is reasonable to think that this structure can be dropped when relying on transformers for parameterizing flow or diffusion models.
+ The evaluation on ImageNet benchmark shows better results for the proposed SF-SiT, compared to presented baselines, on a fixed training budget of 400k iterations and when sampling with an Euler ODE solver. Besides obtaining improved gFID results, sampling requires almost 4 times less FLOPs, thanks to the increased latent compression rate.

Weaknesses:
- This is not the first work introducing the concept of 1D image (sparse) tokenization.
For instance, [B] develops a highly similar idea, presenting a transformer-based 1D tokenizer for sparse tokenization (e.g., 32 tokens) and a decoder that is also based on a ViT and directly decodes 1D tokens into an image.
Indeed, a comparison with the "TiTok" approach [B] or other immediate follow-up works is missing. Slot-based auto-encoders [C] (and much follow up work) are, up to my knowledge, yet another way of designing 1D image tokenizers, and hence are also worth a discussion in the related work section at least, or possibly a comparison.
- The paper does not provide a clear motivation of why we need to explicitly model the distribution of RoIs. I think a motivation / ablation against no explicit modeling, or showing an application (if any) of this "disentangled" latent space is missing. This is important also because the shown examples show a rather regular tiling of the image into RoIs, similar to the usual notion of patches.
- It is not clear whether the reduced number of FLOPs at the sampling / denoising stage (i.e., for SF-SiT) is enough to contrast the increase in FLOPs at the decoding stage, mainly introduced by the proposed neural field decoder, which eventually computes N 2D feature maps (to be blended together) from the N input RoIs, conditioned on the corresponding 1D latents. The evaluation of FLOPs for the entire end-to-end generation would give more insights on the advantage of using the proposed approach.
- An initial attempt of efficiently scaling the training budget with REPA shows promising results on class-conditional generation (with and without CFG), although the results are still not on par with baselines. A comparison with SiT-XL using REPA for faster convergence under the same training budget and with CFG is also missing, thus it is hard to infer how good the proposed approach can scale.

Note: My assessment is somewhere between 2 and 3 here. I eventually settled for a 2, since the missing comparison to [B] and follow-up work is a significant oversight, as is the missing discussion of safeguards.

References:
[A] Rombach, R., Blattmann, A., Lorenz, D., Esser, P., & Ommer, B. (2022). High-resolution image synthesis with latent diffusion models. In Proceedings of the IEEE/CVF conference on computer vision and pattern recognition (pp. 10684-10695).
[B] Yu, Q., Weber, M., Deng, X., Shen, X., Cremers, D., & Chen, L. C. (2024). An image is worth 32 tokens for reconstruction and generation. Advances in Neural Information Processing Systems, 37, 128940-128966.
[C] Locatello, F., Weissenborn, D., Unterthiner, T., Mahendran, A., Heigold, G., Uszkoreit, J., ... & Kipf, T. (2020). Object-centric learning with slot attention. Advances in Neural Information Processing Systems, 33, 11525-11538.

---

> ### Author Rebuttal · Authors · 2025-07-31
>
> We really appreciate your time and efforts for detailed reviews and comments on our manuscript. We would like to address concerns as following:
>
> **Q.1. Can the authors compare the proposed approach against other 1D image tokenization strategies? One straightforward example could be the already mentioned "TiTok" [B], which also evaluates on the ImageNet 256x256 benchmark.**
>
> We show the reconstruction results on our methods and TiTok here on ImageNet 256x256 benchmark.
> | method | token | rFID |
> |--------|-------|------|
> | TiTok  | 32    | 2.21 |
> | TiTok  | 64    | 1.71 |
> | TiTok  | 128   | 1.70 |
> | **SF-VAE** | 32    | 1.70 |
> | **SF-VAE** | 64    | 0.70 |
> | **SF-VAE** | 128   | 0.35 |
>
> It is worth noting that the comparison is not strictly fair since our SF-VAE uses continuous latents while TiTok uses discrete latents, so we do not include this comparison in the submission. From results, we can see SF-VAE reaches much better performance compared to TiTok and this is expected to a moderate extent. Importantly, we mainly focus on joint modeling of latents and RoIs in reconstruction and generation, which is not explored in TiTok and other 1D tokenization work. We will add more discussion on the difference between our work and TiTok 1D tokenizations/object centric/slot attention methods  in revised manuscript.
>
> **Q.2. Can the authors compute the total number of FLOPs for sampling an image in pixel space? Which means, also including the necessary operations to convert your 1D tokens into 2D feature maps via the neural field decoder into the total amount of flops.**
>
> We measure the SD-VAE-ft-mae and SF-VAE (64 tokens) decoder's FLOPs. SD-VAE-ft-mae decoder is with 311 GFLOPs and SF-VAE decoder is with 153 GFLOPs. This hugh difference is because we designed SF-VAE decoder, especially pixel decoder, to be light as much as possible, so we configure channel_multipliers to be [1, 1, 2, 4] in SF-VAE decoder while channel_multipliers are [1, 2, 4, 4] in SD-VAE.
>
> We also compute the FLOPs of our "neural field" operations. We use torch.bmm to perform neural field operation:
> ```
> Shape (64, 1024, 64) -> torch.bmm -> Shape (64, 1024, 64) -> Shape (64, 1024, 512) # 64 is the number of tokens, 1024 is (256/8)*(256/8) grid, 64/512 is the channel
> ```
> The result is 5GFLOPs, which is relatively minor to 153 GFLOPs. So overall, SF-SiT-XL needs 29.3 * 50+153 = 1618 GFLOPs to sample a 256x256 image while  SiT-XL needs 114.4 * 50+311 = 6031 GFLOPs (assume 50 sampling steps)
>
>
> **Q.3. Can the authors report (cf. Table 6) the result (gFID) obtained with the SiT-XL baseline using REPA for faster convergence and CFG scale > 0? It will be helpful to really show the potential of the proposed SF-SiT-XL with limited compute.**
>
> Thank you for your advice on that. Till now, there are not publicly-available checkpoints/results of SiT-XL w/ REPA with CFG scale for 1M/1.4M/comparative iterations so we do not report that. If resources available, we might consider reproducing SiT-XL w/ REPA for 1.4M iterations with CFG and report in future version
>
> **Q.4 Can the authors provide more insights on why we need to (explicitly) model the distribution of RoIs? This question is also partially motivated by the fact that the RoIs (cf. Figure 6) seem to flow to a fairly even distribution where each token still corresponds to a small patch of your image. Ablating this choice can be a very good starting point.**
>
> Thank you for the advice. We ablation this choice by designing the SF-SiT to model RoI distributions first (t<0.5) and model latents then (t>0.5). It can be implemented by extending the `async, min` interpolating schedule in Table 3. This means in the inference, we first sample RoIs (analogue to "layouts") from the distribution and when RoIs are completely generated, latents are then sampled. The resulting gFID result is 63.32, much worse than any entry in Table 3 (best 42.0 for beta=2, worst 55.9 for async min), showing the necessity of modeling the distribution of RoIs, especially jointly with latents.
>
> **Q.5. Can the authors provide a more convincing discussion of the issue of safeguarding their image generation approach?**
>
> We are sincerely sorry for missing that point. We did not consider safeguarding our image generation approach back then because our all models are trained on standard ImageNet datasets and ImageNet is a relatively "safe" dataset. I am not an expert on diffusion safety but I would like to say that intuitively, we can safeguard not only latents but RoIs in the SF-SiT pipeline when generating sensitive contents. The safeguarding techniques could be classifier guiding on either latents or RoIs or both. We can also safeguard SF-SiT in a more finer way by detecting a token to contain sensitive details and throwing it away in SF-VAE decoding. We will do a survey on safeguarding and discuss the relevant work and how to integrate these safeguarding methods in future version.

---

> > ### Comment · Reviewer_pbLv · 2025-08-06
> >
> > Dear authors,
> >
> > Thank you very much for your rebuttal. Some of my concerns were cleared up, while others remain. I will try to summarize my findings below:
> > (Q1) The rebuttal does not fully address my concerns, since only rFID (so the reconstruction quality) is reported, but not the more important gFID (generation quality). As the authors mentioned elsewhere, it is not unexpected that the discrete latents lead to worse reconstruction, but it is not clear what the impact is on image generation.
> > As an aside, I am also not convinced by the authors' claim responding to reviewer zU5o that a comparison with TiTok would not be fair. Both approaches propose a 1D tokenization strategy for image generation. Clearly, there are differences, but I believe this comparison is still sensible.
> > (Q2) Thank you for addressing the point convincingly.
> > (Q3) The answer seems not completely convincing. The CFG scale is not a training hyper-parameter, thus obtaining another checkpoint should be not needed to report this result. If results reported in the paper are obtained with a publicly available SiT-XL baseline trained with REPA, tweaking the CFG scale (to report results with CFG scale > 1, e.g., CFG = 1.5) should be possible.
> > (Q4) I found this answer also only partially convincing. The new result shows that joint modeling is necessary and that sampling RoIs first and then latents leads to a worse result. However, I'm not sure whether from this we can infer how useful RoI modeling actually is with respect to only sampling / modeling latents. I believe an ablation against sampling latents only would be needed.
> > (Q5) I appreciate that the authors made some effort to address this issue.
> >
> > Overall, I tend towards keeping my tendency toward rejection. In agreement with reviewer Rnro, I believe the meaning of the RoIs should be explored more. Also, the comparison against existing 1D tokenization approaches is not fully convincing. In light of this, I tend to believe that further revision would be useful before this paper can be published.

---

> > > ### Author Response · Authors · 2025-08-06
> > >
> > > Thank you for your timely comment and constructive feedback. We apologize for the confusion caused in our rebuttal and would like to address your remaining concerns as follows:
> > >
> > > **Q1**
> > >
> > > | method | token | rFID | gFID |
> > > |--------|-------|------|-----|
> > > | TiTok  | 32    | 2.21 | 2.77  |
> > > | TiTok  | 64    | 1.71 | 2.48  |
> > > | TiTok  | 128   | 1.70 | 1.97 |
> > > | SF-VAE/SiT | 64    | 0.70 | 2.76 |
> > >
> > > Your point is right and we greatly value that observation. Discrete latent space generally leads to a worse reconstruction FID but its compact structure eases the generation task and leads to a better generation FID (64 tokens, 2.48 vs 2.76) . Having said that, the gFID performance gap between SF-SiT and TiTok is not particularly large, and this gap might also be due to different generation modeling approaches chosen (masked-based method, MaskGiT, versus diffusion). Importantly, these generation modeling approaches cannot be swapped without substantial changes as one is designed for discrete codes and one is for continuous. Therefore, we believe it might not be very strict to consider this empirical comparison as fair.  We plan to add thorough discussion and comparison to the revised text to clarify that regarding other compression tokenizations, including 1D following up work and slot-based ones, in the related work and experiment.
> > >
> > > While SF-VAE shares some ideas of extreme compression with 1D tokenization methods (e.g., getting rid of 2D grid representation, discussed in submission line 225-227), it is worth noting that our method still differs from 1D tokenization methods a lot, and **it is not appropriate for SF-VAE to be categorized within 1D family**, as it does not operate in a purely 1D space. In both reconstruction and generation modeling, SF-VAE/SF-SiT always explicitly associates each token with **a shiftable RoI in 2D space** and models them jointly, allowing a token to move and attend to its interested region, which is the main novelty and contribution of our work. These 2D RoI representations also maintain the translation equivariance property of this pipeline (given latents, RoIs shifts, visual content shifts accordingly), which 1D tokenization methods cannot have.
> > >
> > > **Q3**
> > >
> > > We apologize for the confusion caused here. We transcribed the result from the original REPA paper (Table 3) to the result of SiT-REPA baseline w/o CFG  with 1M training iters in submission (Table 6), while this 1M-iter SiT-REPA checkpoint is not publicly available (we choose 1M for comparison because our SF-SiT-REPA is trained with 1.4M iters) There is only 4M-iter SiT-REPA checkpoint available and the 4M-iter SiT-REPA baseline is 5.9 w/o CFG and 1.79 w/ best CFG value. We will reproduce SiT-XL w/ REPA trained with 1.4M iters and report results w/ and w/o CFG to enhance clarity.
> > >
> > > **Q4**
> > >
> > > It is worth noting that according to SF-VAE, latent RoIs are different across different images. Indeed, RoIs follow a distribution and we do not know any form of this distribution, so we need to model the sampling procedure and finally sample RoIs. This sampling procedure is inevitable if we use our SF-VAE to extract latents and RoIs.
> > >
> > > If I understand correctly, your concern here is "If all RoI modeling in the pipeline is switched off, that is, RoIs are handcrafted and fixed across different images and across any point in this system (not learnable), how well will SF-VAE/SF-SiT perform?". Then we need to handcraft RoIs in 2D space first, the most safe way would be tiling the RoIs with non-overlapping grids, then, SF-VAE somehow degrades into the conventional 2D grid VAE (if the number of tokens fixed, also similar to DC-AE). The comparison against the conventional 2D grid VAEs including DC-AE is already presented in the submission (Table 1). But this is a good idea and we really appreciate your insight! A 2D grid VAE simulated by SF-VAE w/ handcrafted RoIs is really worth exploring for ablations and we will add experiments on it and add results to Table 1 for comparison.
> > >
> > >
> > > Again, we sincerely thank you for your timely discussion and we really appreciate your invaluable feedback to our manuscript. We really hope this response can clarify some confusion and properly address your concerns. Please do not hesitate to continue this discussion with us if you have further unresolved questions.

---

> ### Author Response · Authors · 2025-08-05
>
> Dear Reviewer,
>
> We would deeply value the opportunity to continue our discussion with you and to confirm whether all your concerns have been fully addressed in the rebuttal. Your comments and insights would be extremely valuable in enhancing our submission, and we sincerely look foward to your continued feedback or any remaining questions you may have. If you have any further questions or require additional clarification, please feel free to let us know at any time.

---

### Official Review · Reviewer_Rnro · 2025-07-02

**Clarity:** 2
**Significance:** 2
**Originality:** 2
**Rating:** 3
**Confidence:** 5

**Summary:**

This paper introduce SF-VAE, which encodes an image into a small number of latent vectors along with learnable Regions of Interest, and SF-SiT, a flow-based transformer that jointly models latent and RoI trajectories during generation. This work achieves competitive image synthesis results onImageNet data, reducing computational cost significantly compared to conventional grid-based methods.

**Questions:**

How does your method perform on dense or structured visual inputs, such as documents, charts, or scenes with many small objects? ImageNet is not sufficient to test this.

What happens when the number of RoIs is insufficient for reconstruction or generation? Are there strategies to adaptively allocate more tokens to difficult regions?

**Ethical Concerns:**

["NO or VERY MINOR ethics concerns only"]

**Final Justification:**

After reviewing the rebuttal and additional experiments, I maintain a borderline reject recommendation. While the paper presents a promising and efficient approach for sparse image generation using learnable RoIs, key concerns remain unresolved. The method shows limited generalization to complex or dense scenes. Additionally, the learned RoIs lack semantic alignment or interpretability, raising questions about the broader applicability and robustness of the proposed representation.

**Limitations:**

The paper does not discuss how the method would generalize to domains with high spatial complexity (e.g., documents or multi-object scenes), nor whether RoIs are semantically meaningful. These are crucial questions given that the method is explicitly designed to model spatial sparsity.

**Paper Formatting Concerns:**

None.

**Quality:**

2

**Strengths And Weaknesses:**

Strenghs:
1. The introduction of RoI-based latent modeling in a diffusion/flow generation pipeline is an  interesting idea.
2. This RoI base mehtod achieves strong FID scores using significantly fewer tokens and FLOPs than conventional models (e.g., DiT, SiT).


Weaknesses
1. All experiments are restricted to image generation (reconstruction or synthesis) on ImageNet, without any assessment on image understanding tasks such as classification, retrieval, or layout prediction. This leaves open whether the learned latent representations are semantically meaningful or generalizable.

2. Furthermore, ImageNet mainly contains single-object, center-biased natural images. These are spatially sparse by design and make it easy for the RoI model to leran by locating obvious foregrounds. As such, the proposed method is not stress-tested on structured, multi-object, or text-rich data where adaptive sparsity is most needed.

3. The paper lacks any attempt to quantify the semantic alignment, consistency, or interpretability of the learned RoIs. It is unclear whether RoIs focus on objects, backgrounds, or arbitrary textures. The authors can apply this learned representation on vision-langauge problems.

---

> ### Author Rebuttal · Authors · 2025-07-31
>
> We would like to thank the reviewer's time and comments on our submission. We address the concerns as following:
>
> **Q1. SF-VAE latent representations understanding capabilities**
>
> It is a great potential direction but differs a lot from this paper's scope and thus we have not yet explored. This paper is mainly for image reconstruction and generation. Variational autoencoders (VAE) for diffusion latent spaces commonly only compress perceptual details into compact latents and do not encode much high-level semantics into latents. We believe SF-VAE latents also follow this property: compress spatial patterns into latents and shiftable RoIs (can be inferred from Fig 3 and Fig 6, RoIs are still based on low-level details). Therefore, this is a very open and challenging topic on understanding capabilities of SF-VAE latents and we may investigate this in the future.
>
> **Q2. SF-VAE on documents, charts, or scenes with many small objects**
>
> We here evaluate the SF-VAE on MS COCO val 5k images (256x256), which commonly consists of many foreground objects. The results are
>
> | method            | rFID | PSNR | SSIM |
> |-------------------|------|------|------|
> | SD-VAE-ft-mse     | 4.70 | 24.5 | 0.71 |
> | SD-VAE-ft-ema     | 4.42 | 23.7 | 0.69 |
> | **SF-VAE, 64 tokens** | 4.73 | 22.9 | 0.75 |
>
> We can see SF-VAE is genereally on par with SD-VAE on natural complex scene MS COCO images with a better SSIM score.
>
> We evaluate our methods on documents and find not-satisfying results (distortion and broken texts), especially small texts. It is mainly due to **a)** training data, as now we only train SF-VAE on ImageNet (texts are very rate in ImageNet training set). SD-VAE-ft-ema also performs poorly on small texts as they are not finetuned on text dataset; **b)** high compression rate. If we lower compression rate of SF-VAE by a large margin, we believe it would be better for high-frequency details. However, increasing latent space is not meaningful for efficient and compact latents for generation, which is the main goal of our paper.
>
>  **Q3. Semantics and interpretability of learned RoIs**
>
> It can be inferred from Fig 3 and Fig 6 that learned RoIs mainly focus on low-level details and are influenced by the pixel frequency within their covered regions. Specifically, RoIs tend to be larger in areas dominated by low-frequency content. There are not many strong semantics in RoIs since we do not impose any semantic alignment loss, and this is expected and pretty common for self-supervised pixel reconstruction task for training to only encode low-level frequency details in RoIs (as discussed in latent diffusion paper).
>
> **Q4. Are there strategies to adaptively allocate more tokens to difficult regions?**
>
> It is worth noting that our SF-VAE is an end-to-end model and RoIs in SF-VAE can move to high frequency areas by interative shifting by SparseFormer encoding. But handcrafting strategy is a great idea! We have not yet explored in that direction on adaptive allocations of tokens, especially when the number of RoIs is insufficient for reconstruction. A potential simple straightforward way is to iteratively auto-encoding an images with increasing token/RoI numbers on the pixel l1 loss/LPIPS criterion. When the good criterion is met, we stop increasing token numbers. This approach requires training a varying token SF-VAE, which is also simple to implement. Thank you for your advice and we hope can future work explore in this direction further.

---

> ### Author Response · Authors · 2025-08-05
>
> Dear Reviewer,
>
> We would greatly appreciate the opportunity to continue our discussion with you and to confirm whether all your concerns have been fully addressed in the rebuttal. Your valuable insights and additional questions would be extremely helpful in further refining our submission, and we sincerely look foward to your continued feedback or any remaining questions you may have. We hope these responses are helpful for your final assessment.

---

> > ### Comment · Reviewer_Rnro · 2025-08-08
> >
> > Thank you for the detailed rebuttal. After carefully reviewing the additional results and clarifications, some of my concerns remain. This paper introduces a sparse latent image model using learnable RoIs for efficient generation, achieving strong results on ImageNet. However, its limited performance on complex or dense scenes such as documents or multi-object layouts reveals a reliance on low-level frequency patterns. The learned RoIs lack semantic alignment or interpretability, which restricts their utility beyond reconstruction-focused tasks.
> >
> > Although the rebuttal includes results on MS COCO, the performance of SF-VAE remains below baseline in both rFID and PSNR, with only a marginal SSIM gain. Furthermore, its limited performance on complex or dense scenes such as documents or multi-object layouts reveals a reliance on low-level frequency patterns. The learned RoIs lack interpretability, which restricts their utility beyond reconstruction-focused tasks.
> >
> > While the method is promising, its current scope and semantic limitations lead me to maintain a borderline reject.

---

> > > ### Author Response · Authors · 2025-08-09
> > >
> > > Dear Reviewer,
> > >
> > > We sincerely appreciate your timely and constructive feedback on our submission. However, as a final note, we would like to clarify and emphasize again that this paper is specifically designed for **efficient reconstruction and generation** rather than **understanding** and our claimed contributions in the submission never go beyond the generative modeling scope. Your concerns regarding the understanding capablities of latents or RoIs beyond reconstruction-focused tasks are indeed insightful but generally applicable to most all reconstruction-based VAEs designed for diffusion generative modeling. While this is certainly an interesting direction for future exploration, it lies beyond the intended focus of our present work.
> > >
> > > Additionally, our SF-VAE training set only includes ImageNet training dataset, so the learned frequency preference towards low level in natural images is expected. Consequently, evaluating the model, which trained solely on natural images, on text-based documents may not provide a fair or insightful assessment.
> > >
> > > We hope this clarifies our intentions and scope of this submission. Again, we truly value the time and effort you have invested in reviewing our work.
> > >
> > > Best,
> > >
> > > The Authors

---

### Comment · Area_Chair_ePKh · 2025-08-03

Dear Reviewers,

Please:

- Read all reviews and author responses carefully;
- Join the discussion, especially if there are any remaining questions;
- Share your first comment as early as you can to allow time for a productive author-reviewer discussion.

Thank you for your responsible reviewing.

Best,

AC

---

### Decision · Program_Chairs · 2025-09-17

**Decision:**

Accept (poster)

**Comment:**

The paper proposes a non-grid latent space with sparse latent vectors and adaptive Regions of Interest (RoIs) for image reconstruction and generation, aiming to improve efficiency. The models show competitive performance while reducing FLOPs compared to grid-based tokenizers and generators.

Reviewers highlighted strengths such as the novelty of non-grid sparse latent modeling, efficiency improvements in both token usage and FLOPs, and a clear technical presentation.

Key concerns centered on the interpretability of RoIs, which seem to capture low-level patterns rather than semantic structures. Reviewers also noted limited comparisons with 1D tokenizers and scaling baselines, as well as missing experimental details in the original submission.

In response, the authors added experiments on MS COCO, provided a FLOPs breakdown, reported gFID comparisons with TiTok, and expanded ablation studies on RoI size and modeling schedules.

Overall, the AC found the paper interesting and inspiring, with a novel contribution on adaptive and sparse image representations that will benefit research in image generation. While the introduced RoI differs from the usual expectation of semantically coherent RoIs, since the paper did not claim to produce semantic RoIs, the AC considered the approach sufficient for the paper’s goals. The authors are encouraged to clarify their definition of RoI.